ecology

depopulation, hierarchical model, rewilding, ridge regression, butterfly, climate change adaptation

**Author for correspondence:**
Keita Fukasawa
e-mails: k.fukasawa37@gmail.com,
fukasawa@nies.go.jp

# Positive and negative effects of land abandonment on butterfly communities revealed by a hierarchical sampling design across climatic regions

Naoki Sugimoto[1], Keita Fukasawa[2], Akio Asahara[3], Minoru Kasada[4,5], Misako Matsuba[2] and Tadashi Miyashita[1]

[1]Graduate School of Agricultural and Life Sciences, University of Tokyo, 1-1-1 Yayoi, Bunkyo-ku, Tokyo 113-8657, Japan
[2]Biodiversity Division, National Institute for Environmental Studies, 16-2 Onogawa, Tsukuba, Ibaraki 305-8506, Japan
[3]Team HEYANEKO, 3-22-18-702 Minami-Urawa, Minami-ku, Saitama, Saitama 336-0017, Japan
[4]Graduate School of Life Sciences, Tohoku University 6-3, Aoba, Sendai, Miyagi 980-8578, Japan
[5]Department of Experimental Limnology, Leibniz-Institute of Freshwater Ecology and Inland Fisheries, Alte Fischerhuette 2, 16775 Stechlin, Germany

KF, 0000-0002-9563-457X; MK, 0000-0001-9023-430X; TM, 0000-0003-0091-3224

Land abandonment may decrease biodiversity but also provides an opportunity for rewilding. It is therefore necessary to identify areas that may benefit from traditional land management practices and those that may benefit from a lack of human intervention. In this study, we conducted comparative field surveys of butterfly occurrence in abandoned and inhabited settlements in 18 regions of diverse climatic zones in Japan to test the hypotheses that species-specific responses to land abandonment correlate with climatic niches and habitat preferences. Hierarchical models that unified species occurrence and habitat preferences revealed that negative responses to land abandonment were associated with species that have cold climatic niches and use open habitats, suggesting that species negatively impacted by land abandonment will decline more due to future climate warming. Maps representing species gains and losses due to land abandonment, which were created from the model estimates, showed similar geographical patterns, but some areas exhibited high species losses relative to gains. Our hierarchical modelling approach was useful for scaling up local-scale effects of land abandonment to a macro-scale assessment, which is crucial to developing spatial conservation strategies in the era of depopulation.

## 1. Introduction

Traditional agricultural landscapes, with high spatial heterogeneity and moderate levels of disturbance, harbour a variety of organisms worldwide [1]. However, recent social changes, such as depopulation and population ageing, have resulted in an increase in abandoned fields [2,3]. While land abandonment is expected to decrease biological diversity at the national scale [4,5], it may also provide an opportunity for rewilding by vegetation succession [6,7]. To establish a long-term, broad-scale strategy for biodiversity conservation, it is necessary to identify areas where restoration by human management is effective for enhancing biodiversity as well as areas where rewilding is more effective.

Many studies have examined the relationship between land abandonment and biodiversity, but most have targeted a particular region within a nation [8,9]. These results cannot necessarily be extrapolated to other regions with

different climates and land cover types because species responses to environmental changes may differ in different climatic zones [10,11]. Moreover, a common approach for testing the effect of land abandonment has been to analyse each species separately or analyse diversity metrics, but such approaches cannot predict the responses of species in a community that are less likely to be detected in a field survey. Meta-analysis may be a promising tool for evaluating the effect of land abandonment [12,13], but it has shortcomings of heterogeneity in study design and inconsistent outcomes [14].

A practical approach to resolve these issues is to collect biodiversity and land-use data that are relevant to the spatial scale of land abandonment, with the data collected from multiple climate zones (ecoregions) using a standardized sampling strategy. This makes it possible to evaluate how the climatic niches or habitat preferences of organisms (i.e. typical habitat types inhabited by particular species) affect the relationship between land abandonment and species occurrence. Historically, abandoned settlements (or deserted villages) have emerged globally due to a variety of causes such as economic and environmental changes and disasters [15–18], and they provide unique opportunities to evaluate the long-term effects of land abandonment and recovery of ecosystems and to compare the effects among different climates. Species-specific sensitivity to land abandonment is determined by ecological traits [19,20] related to habitat preferences. Thus, information on habitat preferences that can be easily obtained from the literature would be useful for estimating the effects of land abandonment on species that are not included in field survey data.

Traditional agricultural landscapes have been altered in Japan with urbanization and land abandonment, but land abandonment has become more serious in recent years because of the country's declining human population. For instance, semi-natural grassland areas have decreased by 87% during the past century [21] and agricultural fields have decreased by 26% [22]. These trends will continue in the future; 30–50% of currently inhabited settlements in rural areas are expected to become uninhabited by 2050 [23]. The abandonment of agricultural landscapes with various characteristics, such as paddy fields, dry crop fields and grasslands, has caused declines in species diversity of various taxa [13,24–28]. As a result, many organisms are expected to become endangered [29], and butterflies are a typical example. In particular, grassland butterfly species are decreasing substantially; they account for approximately 70% of all Red List butterfly species in Japan [30]. These grassland species are elements of the eastern Eurasian temperate steppe biome [31], which is located in a cooler region than the Mediterranean and European steppes [32]. These species were common in Japan in the late Pleistocene, when the climate was much cooler than at present and natural grasslands were predominant [33,34]. Later, in the warm Holocene when forests became predominant, open lands maintained by human activity offered refugia for the Pleistocene relic grassland fauna. Thus, the effect of land abandonment is expected to be more severe for species that favour cold climates, since grassland species in eastern Eurasia are likely to be cold-adapted. Meanwhile, land abandonment may increase forest-dwelling species during the process of vegetation succession. Although relatively few forest-dwelling butterfly species in Japan are designated as threatened, there appears to be a trade-off in butterfly conservation strategies with respect to whether we should manage abandoned land to maintain open habitats by human intervention or let succession proceed. Thus, the effect of land abandonment should be evaluated for each species across regions, and spatially explicit planning is needed to optimize conservation strategies. For nationwide evaluations, butterflies are suitable subjects owing to extensive studies of their geographical distributions, preferred habitats and larval host plants [35].

The aim of this study was to clarify the effect of land abandonment on butterfly communities at a national scale, including different climatic regions, and to determine how responses to land abandonment differ among species with different habitat preferences. Here, we chose both inhabited and abandoned settlements in each region across Japan to estimate the effect of land abandonment, defined by inhabitation. We focused on three land-use types, namely dry crop field, paddy field and built-up area (i.e. houses and gardens around them), because they can be identified from past terrain maps and visual inspection in the field. The following hypotheses are addressed in this study: (1) species preferring colder temperatures are more likely to show negative responses to land abandonment and (2) open habitat species (inhabiting grassland/agricultural lands) generally show negative responses to land abandonment, whereas forest-dwelling species exhibit positive responses. Based on the estimated coefficients of land abandonment for butterflies with different habitat preferences, we created a nation-wide map reflecting the risks and benefits of land abandonment for butterfly diversity; this map could provide a guideline for identifying regions that may benefit from continuous human intervention.

## 2. Material and methods

### (a) Study area

We selected 18 regions that covered most parts of mainland Japan (figure 1) and differed in terms of mean annual temperature (MAT) (electronic supplementary material, table S1). The range of MAT in the selected regions was 4.5–15.7°C, which covers most of the MAT range in areas inhabited by humans in mainland Japan (the range of MAT of all the meteorological stations except Mt Fuji was 6.0–18.8°C) [36]. In each region, we selected both abandoned and nearby inhabited settlements in approximately equal number, with a total of 2–5 settlements, including both types. We selected abandoned settlements from databases of abandoned settlements in Japan [37–39] according to the following criteria: (1) the settlement was inhabited in the past but uninhabited at present; (2) agricultural land was present before abandonment; and (3) the year of abandonment was known. As a result, we surveyed 34 abandoned and 30 inhabited settlements across all regions in Japan. The time elapsed since abandonment varied among settlements, ranging from 8 to 53 years (electronic supplementary material, table S1). Pearson's correlation coefficient between years since abandonment and MAT was −0.16 (95% CI: −0.39, 0.09), and the difference in MAT between abandoned and inhabited settlements was not significant ($p = 0.44$, Student's $t$-test). The mean areas (± standard error) of the abandoned and inhabited settlements were, respectively, $10.6 \pm 3.1$ ha and $12.2 \pm 4.4$ ha, and these areas did not differ significantly.

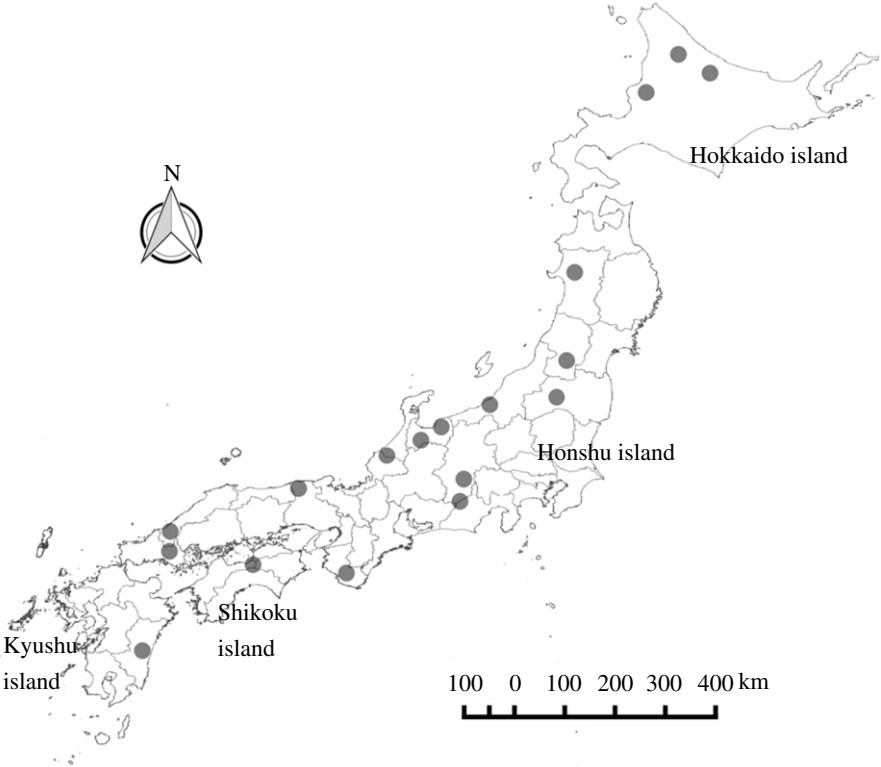

**Figure 1.** Location of study sites. Each site contained both rural abandoned villages and inhabited villages.

## (b) Butterfly survey

We visited each settlement during late May to early August in 2015 and in 2016. To account for differences in butterfly phenology among regions with different temperatures, the visits started in the southern region and ended in the northern region in each year. In 2016, we revisited 16 settlements which we surveyed in 2015 to examine the effect of survey season on butterfly communities. In each visit, we established 4 to 16 plots (each 100 m$^2$) for the butterfly survey, depending on the area of the settlements. The survey plots were located at dry crop fields, paddy fields and built-up areas, and the land use category at each plot was also recorded. All the survey plots in inhabited settlements were under active land use; crop fields and paddy fields were cultivated, and houses were occupied. By contrast, in the abandoned settlements, all human activity was absent and some houses had even collapsed. The butterfly survey was conducted on sunny or slightly cloudy days from 8.00 am to 12.30 pm. A 5-minute census was conducted from roads or rural paths adjacent to the survey plots, and the presence/absence of all species of butterflies found at the plot was recorded. As we did not capture individual butterflies, but instead recorded individuals by sight, some groups of species were difficult to identify to the species level during flight. Thus, the following taxonomic groups were omitted from further analysis: (1) four species of *Papilio* (*P. protenor*, *P. macilentus*, *P. memnon* and *P. helenus*); (2) six species of the tribe Argynnini (*Fabriciana adippe*, *Speyeria aglaja*, *Nephargynnis anadyomene*, *Argynnis paphia*, *Argyreus hyperbius* and *Damora sagana*); (3) Hesperiidae; and (4) three genera of the tribe Theclini (*Neozephyrus*, *Favonius* and *Chrysozephyrus*). We also excluded the non-native species *Pieris rapae* [40] and *P. brassicae*.

The habitat preferences for each butterfly species were determined based on the habitat categorization established by the Japan Butterfly Conservation Society [35]. The categorization consists of the following 11 types: forests, forest edges, open forests, grasslands, crop fields, gardens, built-up areas, rivers, wetland, alpine and rocky habitats. The habitat preferences of a species often comprised several habitat types (electronic supplementary material, table S1).

## (c) Statistical analysis

We applied three statistical approaches: (1) evaluating the correlation between species-specific responses to temperature and land abandonment; (2) evaluating the relationship between habitat preferences of species and susceptibility to land abandonment; and (3) developing a predictive model for species-specific responses to land abandonment based on habitat preferences. Our dataset had a hierarchical structure in which settlements were nested within a region, and a settlement contained multiple land-use types. Survey plots with the same land-use type were generally spatially aggregated in a settlement. Therefore, we incorporated region-, settlement- and land-use-type-level random effects into the analyses to accommodate the spatially hierarchical sampling design. Then, we treated the presence/absence of a species in a plot as the unit of analysis. These models were implemented by a hierarchical Bayesian approach.

### (i) Correlation between the species-specific response to temperature and land abandonment

We applied generalized linear mixed models (GLMMs) [41], incorporating heterogeneous responses to environmental properties among species as a random slope to determine whether species that prefer colder temperatures are more likely to respond negatively to land abandonment. The presence and absence of the $i$th species at the $j$th survey plot, $Y_{ij}$, was assumed to follow a Bernoulli distribution, Bernoulli ($Y_{ij}$; $p_{ij}$), where $p_{ij}$ is the probability of occurrence. The parameter $p_{ij}$ was assumed to be expressed by a logistic regression model. We used land abandonment ($ABAN_j$) and mean annual temperature ($TEMP_j$) as explanatory variables. We also included month surveyed ($MONTH_j$) and a categorical variable of three land-use types ($LU_j$: dry field, built-up area and paddy field [baseline category]) as confounding factors. $TEMP_j$ was obtained from the Climate Mesh Data 2000 [42]. $MONTH_j$ and $LU_j$ were confounding factors controlling for the possible influence on the other parameters of seasonality and land-use type at each survey plot, respectively. We also considered region-, settlement- and land-

use-type-level random effects, namely $\varepsilon_{reg(j)}$, $\varepsilon_{set(j)}$ and $\varepsilon_{lu(j)}$, following normal priors with mean zero and variance $\sigma_r^2$, $\sigma_s^2$ and $\sigma_l^2$, respectively, to accommodate the hierarchical sampling scheme with unbalanced sample size [43]. Overall, $p_{ij}$ was modelled as follows:

$$\begin{aligned} \text{logit}(p_{ij}) = {} & \beta_{0i} + \beta_{1i}ABAN_j + \beta_{2i}TEMP_j + \beta_{3i}MONTH_j \\ & + \beta_{4iI}(LU_j = \text{dryfield}) + \beta_{5i}I(LU_j = \text{built} - \text{up area}) + \varepsilon_{reg}(j) \\ & + \varepsilon_{set}(j)\varepsilon lu(j), \end{aligned}$$
(2.1)

where $\beta_{0i}$ is a species-specific fixed intercept, and $\beta_{1i}$, $\beta_{2i}$, $\beta_{3i}$, $\beta_{4i}$ and $\beta_{5i}$ are regression coefficients for the $i$th species. $I()$ is an indicator function for the dummy variable $LU_j$.

To infer species-specific effects of land abandonment and the correlation with their preferred temperature, a GLMM with a random slope was applied to estimate regression coefficients for each species. In a random slope model, regression coefficients of species are treated as random effects subject to the same prior distribution. As a prior distribution of $\boldsymbol{\beta}_i = (\beta_{1i}, \beta_{2i})$, the bivariate normal distribution MVN($\boldsymbol{\beta}_i$; $\boldsymbol{\mu_\beta}$, $\boldsymbol{\Sigma_\beta}$) was applied. $\boldsymbol{\mu_\beta} = (\mu_{\beta1}, \mu_{\beta2})$ and $\boldsymbol{\Sigma_\beta}$ are the mean vector and covariance matrix, respectively. The covariance matrix was decomposed to standard deviations and Pearson's correlation coefficient as follows:

$$\Sigma_\beta = \begin{bmatrix} \sigma_{\beta1}^2 & \rho\sigma_{\beta1}\sigma_{\beta2} \\ \rho\sigma_{\beta1}\sigma_{\beta2} & \sigma_{\beta2}^2 \end{bmatrix}$$

Here, $\rho$ is Pearson's correlation coefficient between $\beta_{1i}$ and $\beta_{2i}$; a positive value indicates that a species preferring cooler temperatures is more likely to be negatively impacted by land abandonment. The priors of regression coefficients of confounding factors, namely $\beta_{3i}$, $\beta_{4i}$ and $\beta_{5i}$, followed normal distributions with mean $\mu_{\beta3}$, $\mu_{\beta4}$ and $\mu_{\beta5}$ and variance $\sigma_{\beta3}$, $\sigma_{\beta4}$ and $\sigma_{\beta5}$, respectively.

We calculated posterior distributions of the model by the Bayesian method using vague or weakly informative priors for the hyperparameters of the model. For $\beta_{0i}$, $\mu_{\beta1}$, $\mu_{\beta2}$, $\mu_{\beta3}$, $\mu_{\beta4}$ and $\mu_{\beta5}$, a normal prior with mean 0 and variance 100 was used. A half-Cauchy distribution with a scale parameter set to 5 was used as a weakly informative prior for the standard deviations: $\sigma_r$, $\sigma_s$, $\sigma_{\beta1}$, $\sigma_{\beta2}$, $\sigma_{\beta3}$, $\sigma_{\beta4}$ and $\sigma_{\beta5}$ [44,45]. A uniform distribution with range (−1, 1) was used for the prior of the correlation coefficient, $\rho$. Samples from the posterior distribution were obtained using the No-U-Turn Sampler implemented in RStan 2.21.1 [46] (three chains, 1000 iterations after 1000 burn-in iterations with no thinning). We obtained 3000 posterior samples. Convergence of the Markov chain Monte Carlo (MCMC) algorithm was evaluated using $\hat{R}$ [47], and we adopted $\hat{R} < 1.1$ as a threshold of successful convergence [48]. We also evaluated the goodness of fit by using the posterior predictive $p$-value [49] with the following summary statistics: (1) proportion of presence observations over the dataset; (2) proportion of presence observations at abandoned settlements; (3) proportion of presence observations at inhabited settlements; (4) the Gini coefficient of the proportion of presence observations among the settlements; and (5) the Gini coefficient of the proportion of presence observations among species.

For the explanatory variable $ABAN_j$, we considered two candidate variables, 'abandoned or not' and 'years since abandonment', because it is unknown whether the response is immediate or gradual. We compared the performance of models using 'abandoned or not' and 'years since abandonment' on the basis of the widely applicable Bayesian information criterion (WBIC) [50]. WBIC is a generalization of the Bayesian information criterion which is applicable to both regular and singular statistical models; it asymptotically approximates the negative logarithm of the marginal likelihood. The Bayes factor (the ratio of posterior probabilities of competing models) was calculated as the exponential of the difference in WBICs. The abandonment variable with a lower WBIC was used for further analyses.

## (ii) Relationship between habitat preferences of species and responses to land abandonment

To test whether species with different habitat preferences respond differently to land abandonment, we applied a hierarchical model incorporating a hierarchy of habitat type and species [51]. The model is an extension of the species-level model described in section (1), in which the average effect of land abandonment $\mu_{\beta1}$ was modelled by the linear predictor of habitat preferences as follows:

$$\mu_{\beta1i} = \alpha_0 + \alpha_1 H_{ik},$$
(2.2)

where $H_{ik}$ is a binary variable indicating whether the $i$th species uses the $k$th ($k = 1, 2, \ldots, 11$) habitat type or not. $\alpha_0$ is the intercept, and $\alpha_1$ indicates the difference in the average effects of land abandonment between habitat types, where positive and negative values of $\alpha_1$ indicate positive and negative effects for species using the $k$th habitat type relative to species using other habitat types. The same structure was used to determine the effect of $TEMP_j$; that is, $\mu_{\beta2i} = \alpha_{T0} + \alpha_{T1}H_{ik}$. As in the model of §2c(i) above, slopes for each species vary following a multivariate normal distribution with mean vector ($\mu_{\beta1i}$, $\mu_{\beta2i}$).

We calculated posteriors of $\alpha_0$, $\alpha_1$, $\alpha_{T0}$ and $\alpha_{T1}$ separately for the 11 habitat types. We used vague priors of a normal distribution with mean 0 and variance 1000 for these parameters. Other prior settings were the same as those for the species-level model (§2c(i)). Samples from the posterior distribution were obtained using RStan v. 2.21.1 [46], with the same MCMC settings and posterior diagnostics as in §2c(i).

## (iii) Predictive model of species-specific responses to land abandonment

A hierarchical modelling approach is a powerful tool for predicting species-specific responses to environmental changes [52], and allows us to evaluate the broad-scale distribution of a species. Here, we developed a predictive model of species-specific responses to land abandonment based on the habitat preferences of a species and projected the expected loss and gain of butterfly species. The detailed methods are shown in electronic supplementary material, appendix S1. The model was a multivariate extension of equation (2.2), which includes all the habitat types as linear predictors with a ridge regularization [53] on the coefficients of habitat types. To make projection maps for the effect of land abandonment on butterfly assemblages, we defined the loss (CL) and gain (CG) in butterfly species due to land abandonment, which were calculated from the output of the predictive model. The CL and CG are the sum of the predicted negative and positive effect of land abandonment on species present in a spatial unit, respectively. We used nationwide range maps of 70 butterfly species in Japan (electronic supplementary material, table S3), with resolution of approximately 1 km, which were projected by Kasada et al. [54] using distribution models (MaxEnt) applied to butterfly records from a national survey of butterfly distribution in Japan. We evaluated these indices for all the species and for the Red List species [55] separately.

## 3. Results

A total of 49 butterfly species were recorded during our field surveys (electronic supplementary material, table S2). The top 10 species with the highest average occurrence (i.e. number of

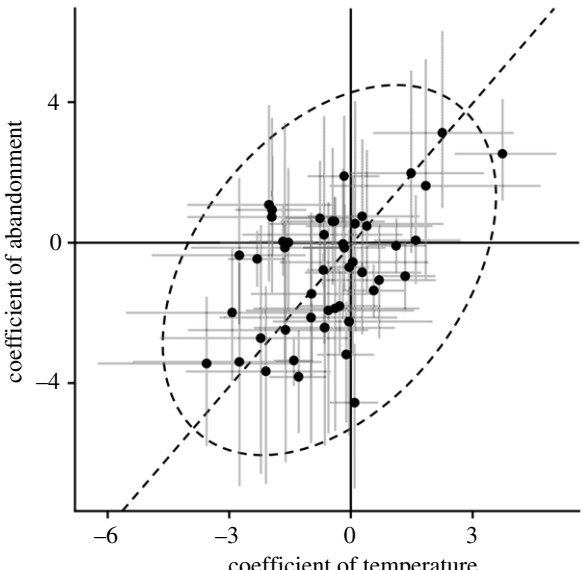

**Figure 2.** Relationship between coefficients of mean annual temperature and land abandonment. Black dots and grey error bars indicate posterior means and 95% credible intervals of species, respectively. The dashed ellipsoid and line are 95% equiprobable ellipsoid and principal axis of coefficients.

presence plots divided by total number of plots) were *Pieris melete, Colias erate, Celastrina argiolus, Ypthima argus, Libythea celtis, Plebejus argus, Lycaena phlaeas, Eurema mandarina, Papilio machaon* and *Apatura metis*, in decreasing order, and they represented 73.4% of the total occurrence records.

The parameters of all models successfully converged ($\hat{R} < 1.1$). In posterior predictive checking, no summary statistics of the observed data showed significant divergence from the posterior predictive distributions generated from all estimated models (electronic supplementary material, table S4). Model selection using the WBIC indicated that the model including 'abandoned or not' (WBIC = 3577.68) outperformed the model for 'years since abandonment' (WBIC = 3611.28), suggesting relatively immediate responses to land abandonment. The Bayes factor for the former model against the latter was $3.89 \times 10^{14}$. Among the 49 butterfly species analysed, 13 showed a significant negative response (i.e. the upper 95% CI of the regression coefficient of 'abandoned or not' was less than zero), while 3 species showed a positive response (electronic supplementary material, table S5). *Lycaena phlaeas, Papilio machaon* and *Papilio xuthus* exhibited the largest negative effect of land abandonment. The three species that showed positive responses were *Limenitis camilla, Cyrestis thyodamas* and *Graphium sarpedon*. Effects of land abandonment and temperature were positively correlated (figure 2); Pearson's correlation coefficient between the effect of land abandonment and annual mean temperature was 0.4 with a 95% CI of (0.07, 0.7).

There was a significant relationship between habitat types of species and responses to land abandonment for four habitat types; species using farmland, grassland and built-up areas responded negatively. Although those using forest showed a significant difference from the other habitat types, the 95% CI of the average response to land abandonment overlapped with zero (figure 3; electronic supplementary material, table S6).

Using a hierarchical model with ridge regularization, we obtained the predictive model for the effect of land abandonment based on the habitat preferences of each species

(electronic supplementary material, table S7). Although the parameter uncertainty of each regression coefficient obtained by ridge regression was large due to the correlation between habitat preferences, the model retained discriminative ability for species that respond negatively to land abandonment (electronic supplementary material, figure S1). By applying this model, posterior means of 37 and 33 species were negative and positive, respectively, and the upper 95% CIs of seven species were negative. Using the predicted values thus obtained, we drew maps of CL (figure 4a) and CG (figure 4b) of species throughout Japan's mainland. The geographical distributions of loss and gain were similar on a coarse scale and the Pearson's correlation coefficient was 0.55; specifically, larger values of both CL and CG were found in inland central Kyushu, inland western Honshu, inland central Honshu, northern Honshu and Hokkaido, while smaller values were found in high-elevation areas. However, CL outweighed CG in 99.6% of grid cells (358 045 of 359 429) and CL tended to be higher in lowlands (figure 4c–e). Standard deviations of CL and CG and the proportion of CG (electronic supplementary material, figure S2a, b,d) were almost directly proportional to their posterior means (Pearson's correlation coefficient $r = 0.86$, 0.99 and 0.96, respectively). The standard deviations of the proportion of CL and CG – CL (electronic supplementary material, figure S2c,e) had stronger positive correlations with CG ($r = 0.87$ and 0.95, respectively) than did their posterior means ($r = -0.96$ and 0.51), reflecting the large uncertainty of CG.

Of the Red List species, 10 were evaluated as victims of land abandonment and 4 were evaluated as beneficiaries. Distribution patterns of CL and CG for Red List species of Japan are shown in electronic supplementary material, figure S3a,b. The correlation between CL and CG was weaker for Red List species than for all species ($r = 0.36$), and their spatial patterns were different on a coarse scale. For instance, CL was small while CG was large in northern Honshu. Conversely, CL was large but CG was small in eastern Hokkaido (electronic supplementary material, figure S3c, d,e). Similar to the maps for all species, the standard deviations of the CL and CG and the proportion of CG (electronic supplementary material, figure S4a,b,d) were almost directly proportional to their posterior means ($r = 0.95$, 0.99 and 0.92, respectively), and the standard deviations of the proportion of CL and CG – CL, rather than their posterior means, were correlated with CG (electronic supplementary material, figure S4c,e).

## 4. Discussion

As expected, the responses of butterflies to land abandonment were associated with the climatic niches and habitat preferences of species. Through our approach in this study, which compared community assemblages between inhabited and abandoned areas using a standard survey protocol, we were able to estimate the effects of land abandonment over a broad range of climatic conditions. In combination with the hierarchical modelling approach, we were able to draw nationwide potential maps of the loss and gain of species richness through land abandonment.

The legacies of the past climate and past land use can affect current functional species compositions within regional species pools [56,57], and different functional compositions

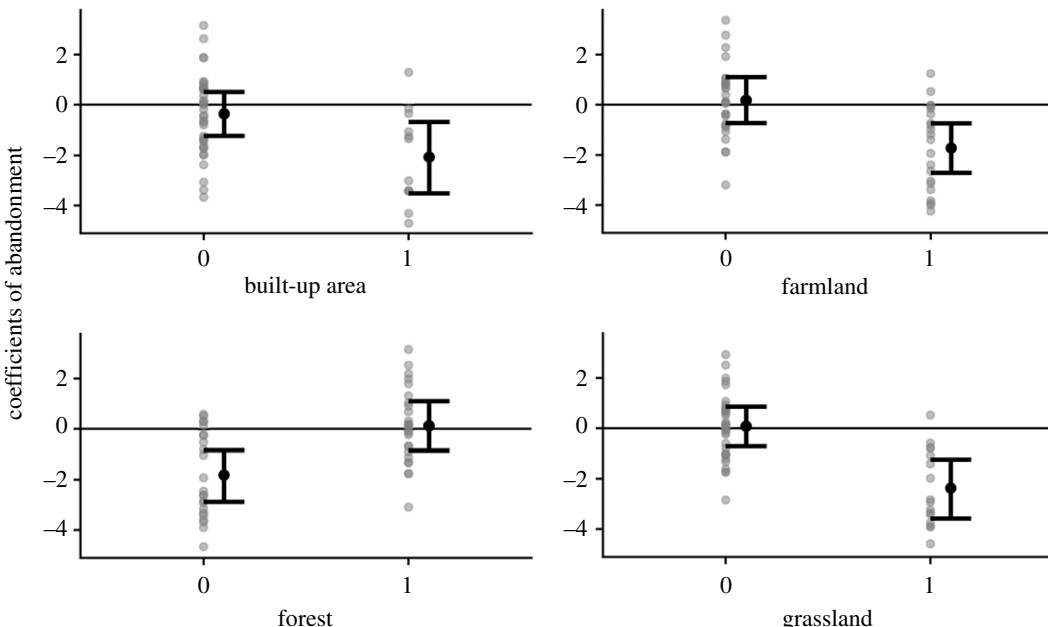

**Figure 3.** Effects of land abandonment on the occurrence of butterflies with different habitat characteristics. On the x-axis, 0 indicates that the species does not use the habitat type and 1 indicates that it does. Grey dots represent coefficients of abandonment ($\beta_{1i}$) for each species. Black dots and error bars represent posterior means and 95% credible interval (95% CI) of the expected value of the coefficient of abandonment ($\mu\beta_{1i}$ in equation (2.1)), respectively. Four habitat types with significant effect on $\mu\beta_{1i}$ (i.e. 95% CI of $\alpha_1$ in equation (2.2) did not overlap 0) were shown.

among regions can result in different community-level responses to current environmental changes [58]. Such a difference would also reflect the historical background of organisms inhabiting seminatural environments in Japan. Grassland was more abundant in the Pleistocene because the climate was colder and drier than at present [34]. After the last glacial period, open lands created by human activity have offered refugia for the Pleistocene relics [59], and such refugia are unlikely to be sustainable if they are abandoned [60]. Our results also raise a conservational concern that species negatively impacted by land abandonment will decline more due to climate warming in the future, because the species susceptible to land abandonment tend to prefer low temperatures (figure 2). This means that continuing land management in rural areas would be an effective measure of climate change adaptation for biodiversity conservation in Japan.

Habitat degradation for open-land butterflies due to land abandonment occurred quickly, whereas forest butterflies would require a longer period of time to experience a positive effect. We suggest that land abandonment resulted in a decrease in the richness of herbaceous plant species [24], leading to the loss or decline of host plants for open habitat butterflies (grassland, crop field and residential species). Indeed, we often observed tall grasses, such as *Miscanthus sinensis*, as predominant species in abandoned paddies, dry fields and built-up areas in uninhabited settlements. The monodominance of wind-pollinated *Miscanthus* would decrease the abundance and diversity of nectar-producing flowers that are important for adult butterflies. By contrast, forest species tended to increase in response to land abandonment, although the proportion of such species was low. Development of old-growth forest in Japan requires approximately 150 years [61], and some tree species require more than a hundred years to recover [62]; accordingly, recovery of forest butterfly species that depend on such trees would also require very long time as well.

It should be noted that there was a positive correlation between losses and gains in butterfly richness both for all species and for Red List species due to land abandonment at the level of grid cells, indicating a trade-off between the benefits of active management of abandoned sites and letting succession proceed. As threatened butterflies in Japan include more grassland species than forest species, human intervention should be prioritized, in principle, to conserve such species, but deliberate decision-making is required for areas where the benefit of land abandonment is high. Our results indicate that conservation of butterflies by active management is desirable in lowland and eastern Hokkaido, where the loss of species that prefer abandoned landscapes would be smaller under such management. Highly cost-effective areas for habitat management based on species ranges and habitat preferences need to be identified with high resolution, because maintaining secondary natural environments is generally labour-intensive [63]. The maps of gain and loss of butterfly diversity presented in this study did not incorporate spatial heterogeneity in land abandonment and instead assumed that all locations in Japan were abandoned. In actuality, the current human population density and the speed of decline is spatially heterogeneous, which will create heterogeneous land-use patterns in the future [64,65]. However, our predictions would still be useful for evaluating biodiversity loss and gain in rural landscapes because human population decline is expected to be severe in most of those areas.

Here, we evaluated the risks and benefits of land abandonment using butterfly occurrence data; however, it is also important to clarify how the abundance of larval host plants and adult nectar plants changes over time after abandonment. Previous studies have shown that many grassland species are still found even 10 years after abandonment [66], indicating the presence of host and nectar plants during that period. In our study, the effects of land abandonment did not show a clear increase over time, as the model with a

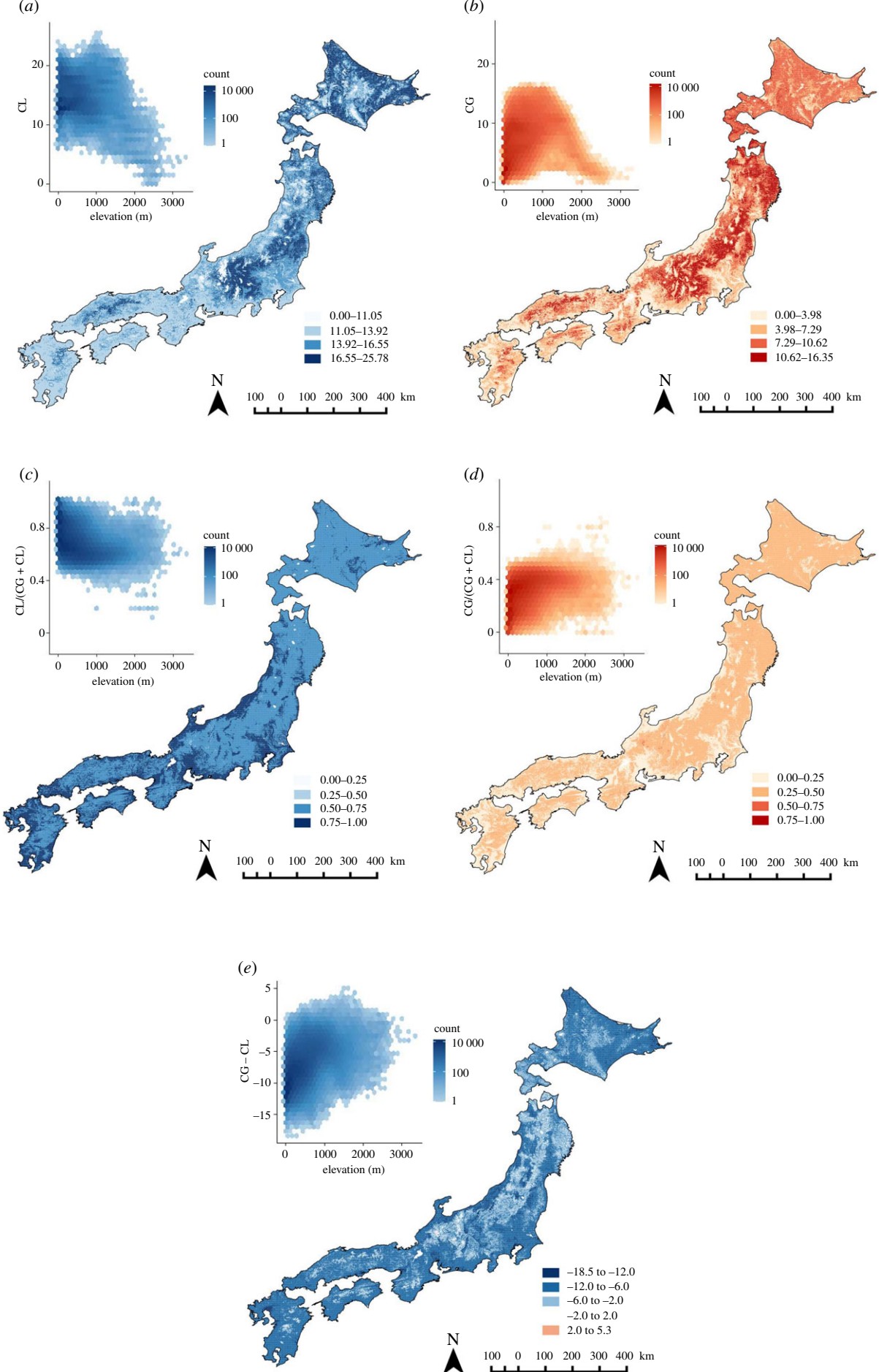

**Figure 4.** Maps of (*a*) cumulative loss (CL), (*b*) gain (CG), proportions of (*c*) CL and (*d*) CG, and (*e*) the difference between CG and CL (CG − CL) in butterfly species richness due to land abandonment for all the species. The boxed graphs are hexagonal binning plots showing their relationship with elevation. (Online version in colour.)

dichotomous variable (inhabited versus abandoned settlements) outperformed the model with a continuous variable (years since abandonment). As Japan has a monsoon climate, with relatively high temperatures and rainfall in comparison with Europe, vegetation succession may proceed within a few years, resulting in a rapid response to land abandonment. This suggests the need to restore habitats as soon as possible after abandonment. The potential maps of the effect of land abandonment will contribute to prioritizing habitat management plans in areas where settlement abandonment in the immediate future and a large decrease in biodiversity after abandonment are expected.

A major contribution of our study was the ability to apply the estimated relationship between habitat preferences and responses to land abandonment to butterfly species that were not observed in the field survey. Potential drawbacks of our survey design are sparse occurrence records and low completeness of species in the dataset due to the small amount of survey effort per sample. Hierarchical modelling could overcome the problems of sparse data, such as model unidentifiability, by introducing a hierarchy of superspecies that stabilize the species-level estimates [67]. With this hierarchical structure, we could predict the responses of butterfly communities even when not all species were well-detected in the field survey. The utility of such a hierarchical modelling approach is widely recognized [68], but there are some caveats, as responses of habitat generalists may be difficult to predict. To resolve this problem, models incorporating responses of host plants to land abandonment and resultant butterfly responses [69] should be explicitly constructed, as has been done for climate change prediction models [70].

Our hierarchical modelling approach also allowed us to evaluate the correlation between susceptibility to climate warming and land abandonment, which could provide important information for the management of semi-natural habitats in order to ensure biodiversity under climate change. However, we should be cautious about extrapolating our results to other regions of in the world with different biogeographical factors. For example, some grassland butterflies in Europe have expanded their range northward in response to the warming climate [71], while such a range expansion is not expected in Japan. Such a difference would arise from the different latitudes of stable grassland biomes. Grassland butterflies in Japan have high commonality with Eurasian temperate steppe fauna ranging from eastern Eurasia to the Mediterranean region, and seminatural grassland in Japan can be thought as the extrazonal remnant of the temperate stable grassland [31]. Interestingly, the climatic limit of eastern Eurasian steppe is much cooler than the Mediterranean region [32]. Europe and Japan are respectively located in the northern and southern regions of the outer part of the Eurasian temperate steppe. Thus, a northward biome shift due to climate warming can help grassland butterflies in Europe, but

not in Japan. A continental-scale comparative study of the responses of grassland butterflies to climate warming is a potential topic for future research to test the hypothesis.

Although we focused on the effect of land abandonment in this study, the hierarchical modelling approach would also be applicable to predicting the response of biological community to climate change if climatic niche information of species [72] is available. A database of climatic niche information of butterflies in Japan has not yet been developed due to the difficulty in collecting the distribution data of butterflies which are widespread in Asia, and this should be a topic for further research.

In conclusion, a hierarchical modelling approach that incorporates the relationship between the habitat preferences of species and their responses to land abandonment was useful for scaling up the results of comparative field studies to a macro-scale evaluation of species losses and gains due to land abandonment. Land abandonment had both positive and negative effects on different functional groups of butterflies in Japan, and our approach could be used to identify areas exhibiting high species losses relative to gains after land abandonment. We believe this approach is promising for large-scale and comprehensive biodiversity assessments regarding the risks and benefits of land abandonment, which are urgently needed in view of ongoing depopulation in developed countries.

Ethics. Our field survey was conducted from public roads and rural paths, and no permission was required in compliance with Japanese law. All information of site locations is coarse enough to protect landowners' privacy. To avoid a negative impact on conservation due to specimen collection, we have not released specific records of endangered species.

Data accessibility. The datasets, program codes and model predictions shown in figure 4 and electronic supplementary material, figures S2, S3 and S4 are available from the Dryad Digital Repository (https://doi.org/10.5061/dryad.6hdr7sr0z [73]), except specific records of endangered species.

The data are provided in electronic supplementary material [74].

Authors' contributions. N.S.: conceptualization, data curation, formal analysis, investigation, resources, software, visualization, writing—original draft; K.F.: conceptualization, data curation, formal analysis, funding acquisition, methodology, project administration, software, supervision, visualization, writing—original draft, writing—review and editing; A.A.: resources; M.K.: resources; M.M.: resources; T.M.: conceptualization, funding acquisition, methodology, project administration, resources, supervision, writing—original draft, writing—review and editing.

All authors gave final approval for publication and agreed to be held accountable for the work performed therein.

Competing interests. We have no competing interests.

Funding. This work was supported in part by the Environment Research and Technology Development Fund (S9) of the Ministry of the Environment, Japan, and JSPS KAKENHI grant nos. 16K16223 and 21H03656.

Acknowledgement. We thank Dr Oliver Schweiger and an anonymous reviewer for providing valuable comments on the earlier version of our manuscript.

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
