## [Peer Review File · Proceedings of the Royal Society B: Biological Sciences]

Review History

RSPB-2021-1163.R0 (Original submission)

Review form: Reviewer 1 (Rob Cooke)

Recommendation

Accept with minor revision (please list in comments)

Scientific importance: Is the manuscript an original and important contribution to its field?

Good

General interest: Is the paper of sufficient general interest?

Good

Quality of the paper: Is the overall quality of the paper suitable?

Good

Is the length of the paper justified?

Yes

Should the paper be seen by a specialist statistical reviewer?

Yes

Do you have any concerns about statistical analyses in this paper? If so, please specify them explicitly in your report.

No

It is a condition of publication that authors make their supporting data, code and materials available - either as supplementary material or hosted in an external repository. Please rate, if applicable, the supporting data on the following criteria.

Is it accessible?

Yes

Is it clear?

Yes

Is it adequate?

No

Do you have any ethical concerns with this paper?

No

Comments to the Author

The authors investigate the effect of land abandonment on butterfly species across Japan, with strong implications for conservation and land management. Specifically, the manuscript highlights areas where restoration by human management is effective for enhancing biodiversity (areas of high cumulative loss following abandonment), as well as areas where rewilding is more effective (areas of high cumulative gain following abandonment).

Overall, I think this is an interesting and well-executed analysis and manuscript. I have a few potential revisions that I believe could help tidy up the manuscript and make it more useful to practitioners. I have attached a number of minor comments directly to the PDF of the manuscript, and raise one more broad point here:

In the methods the authors state, "Losses and gains of species were estimated separately, rather than as a combined gain-minus-loss, because separate estimation is more informative for decisions regarding conservation practices." While I agree that separate maps of cumulative losses and gains are informative and useful, I also think that it would be very instructive for practitioners to map out the combined gain-minus-loss across Japan (or greater justification for not including a combined map would be required). This would provide greater evidence and support for statements in the abstract and introduction e.g. "It is therefore necessary to identify areas that may benefit [this must surely relate to net benefit] from traditional land management practices and those that may benefit from a lack of human intervention". As well as points made in the Discussion e.g. "and our approach could be used to identify areas exhibiting high species losses relative to gains after land abandonment." (this is currently very difficult to do from the figures presented) and would better underpin the overall aims of the paper to identify areas where human management is beneficial and areas where rewilding is beneficial.

The models constructed appear well founded, but I would like to see much more description/discussion about the quality of the models. For instance, is the data representative (how well do the land abandonment survey sites cover the environmental space of Japan), did the models converge (1,000 iterations with 1,000 burn-in iterations seems relatively low - did the models achieve convergence, what were the Rhats of the models, did you thin the data, how many iterations were sampled from the posterior distribution?), and what is the validity/goodness-of-fit of the models (the authors could potentially include posterior predictive checks to give a guide on the validity of the different models).

Review form: Reviewer 2 (Oliver Schweiger)

Recommendation

Major revision is needed (please make suggestions in comments)

Scientific importance: Is the manuscript an original and important contribution to its field?

Good

General interest: Is the paper of sufficient general interest?

Good

Quality of the paper: Is the overall quality of the paper suitable?

Good

Is the length of the paper justified?

Yes

Should the paper be seen by a specialist statistical reviewer?

No

Do you have any concerns about statistical analyses in this paper? If so, please specify them explicitly in your report.

Yes

It is a condition of publication that authors make their supporting data, code and materials available - either as supplementary material or hosted in an external repository. Please rate, if applicable, the supporting data on the following criteria.

Is it accessible?

Yes

Is it clear?

Yes

Is it adequate?

Yes

Do you have any ethical concerns with this paper?

No

Comments to the Author

In their manuscript the authors address an interesting topic: potential trade-offs caused by positive and negative effects of land abandonment and corresponding rewilding. They use butterfly communities in Japan to identify species-specific responses and habitat preferences together with distribution data to predict and map gains and losses under assumed abandonment. However, I have some major concerns which should be addressed to improve the manuscript.

- i) Habitat requirements are not species traits in a strict sense (measurable on an individual). The authors should name them properly.
- ii) While the responses according to habitat requirements are as expected, e.g. openland species decline under abandonment, an increase of warm loving species under abandonment is surprising. Usually, openland species tend to have warmer niches than forest species. The authors explain this with adaptation of species with cooler niches to Pleistocene grasslands.

However, their conclusion are likely to be biased by the fact that abandonment and temperature are collinear in their data (abandoned areas are much cooler than non-abandoned). Including collinear variables in their analyses can lead to wrong coefficient estimates and even to shifts in the sign. One solution would be to use external information for the temperature niche, e.g. extracted from the distribution data as species temperature index, and use this in a similar manner as the habitat preferences.

iii) Using species 'traits' to predict species-specific responses is a promising tool, but needs thorough validation and estimation of uncertainties. This is currently lacking and should be provided.

iv) The authors should make it more clear that the predictions and maps (Fig. 4) are based on a scenario of full land abandonment across Japan, which is quite unlikely for some areas such as highly productive areas or big towns.

v) Moreover, the scenario focuses on land abandonment only but including scenarios of climate warming, separate and together with abandonment, would make the manuscript even more interesting

vi) In the introduction, the negative effects of increasing intensification and urbanization vs. abandonment are often mixed and should be clearly separated.

vii) Sampling a large number of sites comes at costs of local precision. Five min observation time per location seems a bit short. Further, the authors excluded roughly 50% of the Japanese butterfly fauna. The authors should at least discuss potential drawbacks or explain why this is unlikely to be the case.

Detailed comments

L31-33 This argument seems a bit odd. Cold-adapted grassland species may still suffer from climate warming, even if, or perhaps even because, the land is kept open.

L57 You need to be more precise here. It is possible, e.g. using traits (has been done already) or, e.g. with mechanistic models.

L75 One example where the decrease is very likely not caused by abandonment. Please see general comment.

L83 Perhaps I am confused by my European perspective. Here, many openland species are warm-adapted, while forest species are more cold-adapted. If this is different in Japan, you might explain this more explicitly and better connect to hypothesis 1.

L113 Change to "... (Fig. 1), differing in annual ..."

L119 You may delete point 4. Since this is obvious.

L132 The sampling effort differed among the sites and you should correct for that, e.g. by using the number of 'spots' (you may better call them 'sampling locations' or 'plots') as weights in the analyses.

L133 Please provide more details about the sampling design. Was the proportion of the three land use types sampled the same across the different settlements?

L137 The information about 'spot' size is redundant and can be deleted.

L149-153 It would be good to also use external information about the climatic niche. Please see general comment.

L155 Change to: "We applied three ..."

L159 Please provide more information. Settlement should be nested within region. I guess you used each 'spot' as a data point, or did you aggregate the data to land use type? Please provide more information. Without aggregation, you need to include habitat type as additional random effect, nested within spot, nested within region.

L165 Do you mean you included random slopes and intercepts?

L170 You want to assess large-scale context specificity and I was wondering why you did not include interaction effects of abandonment and temperature. However, be aware that both variables are collinear which can cause wrong coefficient estimates (see general comment).

L172 I would not consider the three land use types as confounding effects but rather as an interesting question itself. Does the impacts of abandonment differ among the land use types? You may test this with respective interaction effects.

L179 What about 'paddy'?

L200 Unclear structure of the sentence.

L219 For me it is not clear how you keep the information about abandonment if you replace it with 'trait' information. Or did you change the response variable using the response to abandonment (respective coefficient from Eqn. 1)?

L290 Change to "... the upper 95% CI ..."

L300 From Fig. 3 it seems that forest species do not respond at all.

L310 Does this mean that 33 species responded positively?

L337 "species pools"

L338 "regions"

L339 "reflect the historical"

L349 Please reorder the sentence from abandonment to herbs to hosts to openland butterflies.

L353ff Good point, but please be more clear to explain that habitat degradation is quite immediate for openland butterflies while positive effects for forest species take much longer.

L363 How did you show and test this?

Fig. 4 Are CL and CG absolute numbers? Please provide proportions instead, which is more informative since it relates to the total number of species predicted in a grid cell.

Best wishes,
Oliver Schweiger

Decision letter (RSPB-2021-1163.R0)

02-Jul-2021

Dear Dr Fukasawa,

I am writing to inform you that your manuscript RSPB-2021-1163 entitled "Hierarchical trait-based model reveals positive and negative effects of land abandonment on butterfly communities across climatic regions in Japan" has, in its current form, been rejected for publication in Proceedings B.

This action has been taken on the advice of referees, who have recommended that substantial revisions are necessary. With this in mind we would be happy to consider a resubmission, provided the comments of the referees are fully addressed. However please note that this is not a provisional acceptance.

- 1) A 'response to referees' document including details of how you have responded to the comments, and the adjustments you have made.
- 2) A clean copy of the manuscript and one with 'tracked changes' indicating your 'response to referees' comments document.
- 3) Line numbers in your main document.
- 4) Data - please see our policies on data sharing to ensure that you are complying (<https://royalsociety.org/journals/authors/author-guidelines/#data>). Please also see the referee's comments on the data required.

Sincerely,
Professor Loeske Kruuk
<mailto:proceedingsb@royalsociety.org>

Associate Editor
Comments to Author:

We received two reviews and both acknowledge that the study is of general interest although both also make a number of suggestions where more detail is needed and arguments should be clarified. Reviewer 2 also has some strong concerns about how the the study is referring to habitat preferences rather than naming the traits in question. There are also some technical questions about co-linearity in the data and its impact and details of the modelling process (no. iterations and burn-in, thinning). I suggest the points raised can be addressed, but also a major revision is required.

Reviewer(s)' Comments to Author:

Referee: 1

Comments to the Author(s)

The authors investigate the effect of land abandonment on butterfly species across Japan, with strong implications for conservation and land management. Specifically, the manuscript highlights areas where restoration by human management is effective for enhancing biodiversity (areas of high cumulative loss following abandonment), as well as areas where rewilding is more effective (areas of high cumulative gain following abandonment).

Overall, I think this is an interesting and well-executed analysis and manuscript. I have a few potential revisions that I believe could help tidy up the manuscript and make it more useful to practitioners. I have attached a number of minor comments directly to the PDF of the manuscript, and raise one more broad point here:

In the methods the authors state, "Losses and gains of species were estimated separately, rather than as a combined gain-minus-loss, because separate estimation is more informative for decisions regarding conservation practices." While I agree that separate maps of cumulative losses and gains are informative and useful, I also think that it would be very instructive for practitioners to map out the combined gain-minus-loss across Japan (or greater justification for not including a combined map would be required). This would provide greater evidence and support for statements in the abstract and introduction e.g. "It is therefore necessary to identify areas that may benefit [this must surely relate to net benefit] from traditional land management practices and those that may benefit from a lack of human intervention". As well as points made in the Discussion e.g. "and our approach could be used to identify areas exhibiting high species losses relative to gains after land abandonment." (this is currently very difficult to do from the figures presented) and would better underpin the overall aims of the paper to identify areas where human management is beneficial and areas where rewilding is beneficial.

The models constructed appear well founded, but I would like to see much more description/discussion about the quality of the models. For instance, is the data representative (how well do the land abandonment survey sites cover the environmental space of Japan), did the models converge (1,000 iterations with 1,000 burn-in iterations seems relatively low - did the models achieve convergence, what were the Rhats of the models, did you thin the data, how many iterations were sampled from the posterior distribution?), and what is the validity/goodness-of-fit of the models (the authors could potentially include posterior predictive checks to give a guide on the validity of the different models).

Additional comments from Rev 1 on data accessibility:

As far as I can tell, only the R code is provided, the data are not included in the Dryad download link. The code mentions a "dataset.csv" but I can't find this at the Dryad link.

I think it'd also be beneficial to include more output data (e.g., the values of CL and CG per grid cell across Japan). These data would likely be useful for practitioners and readers to interpret the results more locally. Some of the output data, e.g., coefficient and parameter estimates, are included in Appendix S2 but it would be good if the spatial results were also made available.

Referee: 2

Comments to the Author(s)

In their manuscript the authors address an interesting topic: potential trade-offs caused by positive and negative effects of land abandonment and corresponding rewilding. They use butterfly communities in Japan to identify species-specific responses and habitat preferences together with distribution data to predict and map gains and losses under assumed abandonment. However, I have some major concerns which should be addressed to improve the manuscript.

i) Habitat requirements are not species traits in a strict sense (measurable on an individual). The authors should name them properly.

ii) While the responses according to habitat requirements are as expected, e.g. openland species decline under abandonment, an increase of warm loving species under abandonment is surprising. Usually, openland species tend to have warmer niches than forest species. The authors explain this with adaptation of species with cooler niches to Pleistocene grasslands. However, their conclusion are likely to be biased by the fact that abandonment and temperature are collinear in their data (abandoned areas are much cooler than non-abandoned). Including collinear variables in their analyses can lead to wrong coefficient estimates and even to shifts in the sign. One solution would be to use external information for the temperature niche, e.g. extracted from the distribution data as species temperature index, and use this in a similar manner as the habitat preferences.

iii) Using species 'traits' to predict species-specific responses is a promising tool, but needs thorough validation and estimation of uncertainties. This is currently lacking and should be provided.

iv) The authors should make it more clear that the predictions and maps (Fig. 4) are based on a scenario of full land abandonment across Japan, which is quite unlikely for some areas such as highly productive areas or big towns.

v) Moreover, the scenario focuses on land abandonment only but including scenarios of climate warming, separate and together with abandonment, would make the manuscript even more interesting

vi) In the introduction, the negative effects of increasing intensification and urbanization vs. abandonment are often mixed and should be clearly separated.

vii) Sampling a large number of sites comes at costs of local precision. Five min observation time per location seems a bit short. Further, the authors excluded roughly 50% of the Japanese butterfly fauna. The authors should at least discuss potential drawbacks or explain why this is unlikely to be the case.

Detailed comments

L31-33 This argument seems a bit odd. Cold-adapted grassland species may still suffer from climate warming, even if, or perhaps even because, the land is kept open.

L57 You need to be more precise here. It is possible, e.g. using traits (has been done already) or, e.g. with mechanistic models.

L75 One example where the decrease is very likely not caused by abandonment. Please see general comment.

L83 Perhaps I am confused by my European perspective. Here, many openland species are warm-adapted, while forest species are more cold-adapted. If this is different in Japan, you might explain this more explicitly and better connect to hypothesis 1.

L113 Change to "... (Fig. 1), differing in annual ..."

L119 You may delete point 4. Since this is obvious.

L132 The sampling effort differed among the sites and you should correct for that, e.g. by using the number of 'spots' (you may better call them 'sampling locations' or 'plots') as weights in the analyses.

L133 Please provide more details about the sampling design. Was the proportion of the three land use types sampled the same across the different settlements?

L137 The information about 'spot' size is redundant and can be deleted.

L149-153 It would be good to also use external information about the climatic niche. Please see general comment.

L155 Change to: "We applied three ..."

L159 Please provide more information. Settlement should be nested within region. I guess you used each 'spot' as a data point, or did you aggregate the data to land use type? Please provide more information. Without aggregation, you need to include habitat type as additional random effect, nested within spot, nested within region.

L165 Do you mean you included random slopes and intercepts?

L170 You want to assess large-scale context specificity and I was wondering why you did not include interaction effects of abandonment and temperature. However, be aware that both variables are collinear which can cause wrong coefficient estimates (see general comment).

L172 I would not consider the three land use types as confounding effects but rather as an interesting question itself. Does the impacts of abandonment differ among the land use types? You may test this with respective interaction effects.

L179 What about 'paddy'?

L200 Unclear structure of the sentence.

L219 For me it is not clear how you keep the information about abandonment if you replace it with 'trait' information. Or did you change the response variable using the response to abandonment (respective coefficient from Eqn. 1)?

L290 Change to "... the upper 95% CI ..."

L300 From Fig. 3 it seems that forest species do not respond at all.

L310 Does this mean that 33 species responded positively?

L337 "species pools"

L338 "regions"

L339 "reflect the historical"

L349 Please reorder the sentence from abandonment to herbs to hosts to openland butterflies.

L353ff Good point, but please be more clear to explain that habitat degradation is quite immediate for openland butterflies while positive effects for forest species take much longer.

L363 How did you show and test this?

Fig. 4 Are CL and CG absolute numbers? Please provide proportions instead, which is more informative since it relates to the total number of species predicted in a grid cell.

Best wishes,
Oliver Schweiger

Author's Response to Decision Letter for (RSPB-2021-1163.R0)

See Appendix A.

RSPB-2021-2222.R0

Review form: Reviewer 2

Recommendation

Accept with minor revision (please list in comments)

Scientific importance: Is the manuscript an original and important contribution to its field?

Excellent

General interest: Is the paper of sufficient general interest?

Excellent

Quality of the paper: Is the overall quality of the paper suitable?

Excellent

Is the length of the paper justified?

Yes

Should the paper be seen by a specialist statistical reviewer?

No

Do you have any concerns about statistical analyses in this paper? If so, please specify them explicitly in your report.

No

It is a condition of publication that authors make their supporting data, code and materials available - either as supplementary material or hosted in an external repository. Please rate, if applicable, the supporting data on the following criteria.

Is it accessible?

Yes

Is it clear?

Yes

Is it adequate?

Yes

Do you have any ethical concerns with this paper?

No

Comments to the Author

The manuscript has been considerably improved and I have only a few minor, mostly editorial comments.

L59 Better change to „that are less likely to be detected“

L240 Did you mean lower WBIC?

L267 Please change to „methods are shown“

L272 „from the output“

L381 „would be smaller“

L414 Please change to „species were well-detected“

L423 There seems something to be missing in „result to other of in the „. Please check.

Best wishes,
Oliver Schweiger

Decision letter (RSPB-2021-2222.R0)

@@date to be populated upon sending@@

Dear Dr Fukasawa

I am pleased to inform you that your manuscript RSPB-2021-2222 entitled "Positive and negative effects of land abandonment on butterfly communities revealed by a hierarchical sampling design across climatic regions" has been accepted for publication in Proceedings B.

The referee and Associate Editor have recommended publication, but have also suggested some minor revisions to your manuscript. Therefore, I invite you to respond to the referee's comments and revise your manuscript. Because the schedule for publication is very tight, it is a condition of publication that you submit the revised version of your manuscript within 7 days. If you do not think you will be able to meet this date please let us know.

- 1) A text file of the manuscript (doc, txt, rtf or tex), including the references, tables (including captions) and figure captions. Please remove any tracked changes from the text before submission. PDF files are not an accepted format for the "Main Document".
- 2) A separate electronic file of each figure (tiff, EPS or print-quality PDF preferred). The format should be produced directly from original creation package, or original software format. PowerPoint files are not accepted.

3) Electronic supplementary material: this should be contained in a separate file and where possible, all ESM should be combined into a single file. All supplementary materials accompanying an accepted article will be treated as in their final form. They will be published alongside the paper on the journal website and posted on the online figshare repository. Files on figshare will be made available approximately one week before the accompanying article so that the supplementary material can be attributed a unique DOI.

Sincerely,

Professor Loeske Kruuk

Associate Editor

Comments to Author:

I agree with the reviewer that the authors did an excellent job revising the original version of the manuscript. The reviewer makes a few suggestions for editorial changes, which should be taken into account. Otherwise I I am happy to recommend the manuscripts for publication in Proc Roy Soc.

Reviewer(s)' Comments to Author:

Referee: 2

Comments to the Author(s).

The manuscript has been considerably improved and I have only a few minor, mostly editorial comments.

L59 Better change to „that are less likely to be detected“

L240 Did you mean lower WBIC?

L267 Please change to „methods are shown“

L272 „from the output“

L381 „would be smaller“

L414 Please change to „species were well-detected“

L423 There seems something to be missing in „result to other of in the „. Please check.

Best wishes,

Oliver Schweiger

Author's Response to Decision Letter for (RSPB-2021-2222.R0)

See Appendix B.

Decision letter (RSPB-2021-2222.R1)

21-Feb-2022

Dear Dr Fukasawa

I am pleased to inform you that your manuscript entitled "Positive and negative effects of land abandonment on butterfly communities revealed by a hierarchical sampling design across climatic regions" has been accepted for publication in Proceedings B.

Data Accessibility section

Open Access

Paper charges

Sincerely,

Proceedings B

Appendix A

Response to the Associate Editor

We received two reviews and both acknowledge that the study is of general interest although both also make a number of suggestions where more detail is needed and arguments should be clarified. Reviewer 2 also has some strong concerns about how the the study is referring to habitat preferences rather than naming the traits in question. There are also some technical questions about co-linearity in the data and its impact and details of the modelling process (no. iterations and burn-in, thinning). I suggest the points raised can be addressed, but also a major revision is required.

Response: We sincerely appreciated the constructive suggestions from the editor and referees.

Throughout the revised manuscript, we have changed the term “habitat traits” to “habitat preferences” (Line 1,2, 29, 35, 39, 64, 72-74, 106, 115, 161, 165, 169, 171, 240, 243, 244, 246, 261, 264, 305, 337, 341, 405, 412, 440-442). In addition, we confirmed that the correlation between land abandonment and climate variables was low enough that multicollinearity did not pose a problem in our statistical analyses (Line 132-134). We have added a more detailed description of the MCMC settings (Line 221-224) and explicitly mentioned that R-hat values were sufficiently low (Line 284). We have also added the results of posterior predictive checks and we could not find any significant discrepancies between the data and model predictions (Line 284-287, Table S4). We fully reanalysed the data considering land-use-level random effects as suggested by Dr. Schweiger, and the results were almost identical to those in the previous version. We have extensively revised the manuscript according to the reviewers’ comments, incorporating their suggestion as much as possible. Our point-by-point responses are provided below.

Please note that we moved a part of the methods and figures to the Appendix to meet the page limit.

Response to Referee 1

I have attached a number of minor comments directly to the PDF of the manuscript

Response: Thank you for these detailed comments. We have revised the manuscript to incorporate the suggestions in the pdf file.

In the methods the authors state, “Losses and gains of species were estimated separately, rather than as a combined gain-minus-loss, because separate estimation is more informative for decisions regarding conservation practices.” While I agree that separate maps of cumulative losses and gains are informative and useful, I also think that it would be very instructive for practitioners to map out the combined gain-minus-loss across Japan (or greater justification for not including a combined map would be required). This would provide greater evidence and support for statements in the abstract and introduction e.g. “It is therefore necessary to identify areas that may benefit [this must surely relate to net benefit] from traditional land management practices and those that may benefit from a lack of human intervention”. As well as points made in the Discussion e.g. “and our approach could be used to identify areas exhibiting high species losses relative to gains after land abandonment.” (this is currently very difficult to do from the figures presented) and would better underpin the overall aims of the paper to identify areas where human management is beneficial and areas where rewilding is beneficial.

Response: We agree that the maps of gain-minus-loss are useful for practitioners to develop spatial conservation plans. In the revised manuscript and supplementary material, we have added the maps of gain-minus-loss (Fig. 4e and Fig. S3e) and their standard deviations (Fig. S2e and Fig. S4e), respectively. We also revised the relevant text in Methods and Results accordingly (Line 45-47 in Appendix 1, 317-327 and 330-334).

The models constructed appear well founded, but I would like to see much more description/discussion about the quality of the models. For instance, is the data representative (how well do the land abandonment survey sites cover the environmental space of Japan), did the models converge (1,000 iterations with 1,000 burn-in iterations seems relatively low - did the models achieve convergence, what were the Rhats of the models, did you thin the data, how many iterations were sampled from the posterior distribution?), and what is the validity/goodness-of-fit of the models (the authors could potentially include posterior predictive checks to give a guide on the validity of the different models).

Response: In the revised manuscript, we have described the extent to which the range of mean annual temperature at our survey sites covers the range in mainland Japan. Mean annual temperature at our

survey sites covered most of the climatic range in Japan except uninhabitable alpine zones (Line 120-124). We also noted that the correlation between land abandonment and climate was so low that we can tease these effects apart with good precision (Line 132-134). In the revised manuscript, we have added more details about posterior sampling (Line 221-222). We have also explicitly mentioned the assessment of R-hat (Line 222-224). We confirmed that the R-hat values for all parameters in all the models were < 1.1 , and we have explained this in the Results (Line 284). The No-U-Turn Sampler implemented in RStan includes many posterior evaluations along a parameter trajectory following Hamiltonian dynamics within an iteration to improve the sampling efficiency of each iteration. With a well-posed model structure and certain tricks such as reparameterization, 1000 iterations with no thinning was sufficient to obtain well-mixed posterior samples. We also conducted posterior predictive checks using posterior predictive p -values for multiple summary statistics (Line 224-229) and confirmed that our dataset did not show significant divergence from the posterior predictive distributions (Line 284-287). We have added a summary table of the posterior predictive p -values in the supplementary material (Table S4).

Additional comments from Rev 1 on data accessibility:

As far as I can tell, only the R code is provided, the data are not included in the Dryad download link. The code mentions a "dataset.csv" but I can't find this at the Dryad link.

Response: As we wrote in the Ethic Statement (Line 450-454), we made the occurrence records of Red List species closed to avoid a negative impact on conservation (more specifically, overexploitation by specimen collectors). The dataset_maskRL.csv on Dryad is a masked version of dataset.csv in which we deleted the presence/absence of red list species. We have explained this more clearly as a comment in the R code. Also, the range maps of butterflies for projection uploaded ("presence.proj_maskRL.csv") is the data without Red List records.

I think it'd also be beneficial to include more output data (e.g., the values of CL and CG per grid cell across Japan). These data would likely be useful for practitioners and readers to interpret the results more locally. Some of the output data, e.g., coefficient and parameter estimates, are included in Appendix S2 but it would be good if the spatial results were also made available.

Response: Prior to resubmission, we uploaded an output file including CLs and CGs with the coordinate information (Line 456-459).

Response to Dr. Schweiger (Referee 2)

i) Habitat requirements are not species traits in a strict sense (measurable on an individual). The authors should name them properly.

Response: Throughout the revised manuscript, we have changed the term “habitat traits” to “habitat preferences” and also revised the title and related sentences to accommodate this change (Line 1,2, 29, 35, 39, 64, 72-74, 106, 115, 161, 165, 169, 171, 240, 243, 244, 246, 261, 264, 305, 337, 341, 405, 412, 440-442).

ii) While the responses according to habitat requirements are as expected, e.g. openland species decline under abandonment, an increase of warm loving species under abandonment is surprising. Usually, openland species tend to have warmer niches than forest species. The authors explain this with adaptation of species with cooler niches to Pleistocene grasslands. However, their conclusion are likely to be biased by the fact that abandonment and temperature are collinear in their data (abandoned areas are much cooler than non-abandoned). Including collinear variables in their analyses can lead to wrong coefficient estimates and even to shifts in the sign. One solution would be to use external information for the temperature niche, e.g. extracted from the distribution data as species temperature index, and use this in a similar manner as the habitat preferences.

Response: The correlation between climate and land abandonment was low in our dataset. Pearson’s correlation coefficient between mean annual temperature (MAT) and years since abandonment was -0.16 (95%CI: $-0.39, 0.09$), and the difference in MAT between abandoned and inhabited settlements was far from significant (p -value = 0.44 , Student’s t -test) (Line 132-134). This was due to our hierarchical sampling scheme that included both inhabited and abandoned settlements within each region. We think the multicollinearity between MAT and abandonment was not a problem in our analyses.

It was striking that climate warming is favourable for grassland butterflies in Europe. Your suggestion was great opportunity for us to reconsider the factors determining the context-dependent effect of land abandonment on community assembly. We considered that the biogeographical characteristics of the Eurasian temperate steppe spanning from north-eastern Asia to the Mediterranean region is the major factor influencing the difference between Europe and Japan. Grassland butterflies in Japan have a high commonality with eastern Eurasian temperate steppe fauna, and seminatural grassland in Japan can be thought as an extrazonal remnant of the temperate stable grassland [1]. Interestingly, the climatic range of the eastern Eurasian steppe is much colder than the Mediterranean steppe [2]. This is due to the difference in relative importance of coldness

and dryness for the formation of stable grassland; in eastern Eurasia, the rainy season occurs during the growing season of plants and winter frost plays a more important role than in the Mediterranean steppe. Europe and Japan are respectively located in the northern and southern regions of the Eurasian temperate steppe. Thus, a northward biome shift due to climate warming will help grassland butterflies in Europe, but not in Japan. To test this hypothesis, further studies are needed to compare the responses of grassland butterflies to climate change across Eurasia.

In the revised manuscript, we have explained that grassland species in eastern Eurasia are likely to be cold-adapted in the Introduction (Line 86-94) and added sentences describing possible factors that account for the different situations in Europe and East Asia (Line 417-433).

iii) Using species 'traits' to predict species-specific responses is a promising tool, but needs thorough validation and estimation of uncertainties. This is currently lacking and should be provided.

Response: We conducted posterior predictive checks to test whether predictions from the estimated models were biased away from the observed data. We confirmed that there was no significant discrepancy between the model predictions and the observed data (Line 284-287, Table S4). We have also shown the 95% CIs of the parameter estimates from the predictive model of the effect of land abandonment (Table S7), the posterior standard error of CL and CG (Fig. S2 and Fig. S4) and the uncertainty of the predicted effect of land abandonment for each species (Fig. S1).

iv) The authors should make it more clear that the predictions and maps (Fig. 4) are based on a scenario of full land abandonment across Japan, which is quite unlikely for some areas such as highly productive areas or big towns.

Response: We have acknowledged that the maps of gain and loss of butterfly diversity correspond to an extreme scenario in which all the locations in Japan were abandoned in the Discussion (Line 382-389).

v) Moreover, the scenario focuses on land abandonment only but including scenarios of climate warming, separate and together with abandonment, would make the manuscript even more interesting

Response: We agree that considering temperature niche information of species is a powerful approach for predicting community-level response to climate change. However, it is difficult to develop a database on climatic niche information (like Schweiger et al. [3]) for Japanese butterflies because the geographical range is so large that occurrence data outside Japan, which is necessary to make precise climatic niche, are currently lacking. We have now acknowledged this and suggested it as a

topic for further research in the Discussion section of the revised manuscript (Line 434-439).

vi) In the introduction, the negative effects of increasing intensification and urbanization vs. abandonment are often mixed and should be clearly separated.

Response: We have removed the topic of urbanization on Line 45. Also, we have revised the sentence on Line 75-77 to emphasize the issue of land abandonment.

vii) Sampling a large number of sites comes at costs of local precision. Five min observation time per location seems a bit short. Further, the authors excluded roughly 50% of the Japanese butterfly fauna. The authors should at least discuss potential drawbacks or explain why this is unlikely to be the case.

Response: We have acknowledged the drawbacks of small per-sample effort, such as sparse occurrence records and low completeness of species in the dataset, and have noted that these drawbacks could be overcome by hierarchical modelling including habitat preferences. With the hierarchical modelling, problems of unidentifiability due to sparseness of data could be avoided. By considering the habitat preferences of species within the hierarchical structure, we could predict the response of butterfly communities even in the case that not all the species were detected in the field survey. (Line 406-412).

L31-33 This argument seems a bit odd. Cold-adapted grassland species may still suffer from climate warming, even if, or perhaps even because, the land is kept open.

Response: As we discussed in our response to the comment ii), Japan is located in a warmer region of the grassland biome in eastern Eurasia, and grassland species tend to be cold adapted. It was difficult to explain this biogeographical factor within a limited space of the abstract, and we have replaced the sentence with the expected consequence of future environmental change (Line 31-32).

L57 You need to be more precise here. It is possible, e.g. using traits (has been done already) or, e.g. with mechanistic models.

Response: We have revised the text as follows to state this more precisely:

“a common approach for testing the effect of land abandonment has been to analyse each species separately or analyse diversity metrics, but such approaches cannot predict the responses of species in a community that are not detected in a field survey.” (Line 55-58)

L75 One example where the decrease is very likely not caused by abandonment. Please see general comment.

Response: As we mentioned above, we have revised the sentence on Line 75-77 to emphasize the issue of land abandonment.

L83 Perhaps I am confused by my European perspective. Here, many openland species are warm-adapted, while forest species are more cold-adapted. If this is different in Japan, you might explain this more explicitly and better connect to hypothesis 1.

Response: We have explained that the grassland species in eastern Eurasia are likely to be cold adapted because the grassland biome in the eastern Eurasia is distributed in a cooler region than Europe, and grassland species in Japan are Pleistocene relics of grassland fauna which was once abundant in a colder climate than the one today (Line 86-94).

L113 Change to "... (Fig. 1), differing in annual ..."

Response: We have revised this as suggested (Line 120-121).

L119 You may delete point 4. Since this is obvious.

Response: We have deleted this point as suggested (Line 129).

L132 The sampling effort differed among the sites and you should correct for that, e.g. by using the number of 'spots' (you may better call them 'sampling locations' or 'plots') as weights in the analyses.

Response: We have replaced all instances of the term "spot" with "plot" in the manuscript. We considered that the use of GLMMs without weighting was sufficient to accommodate the unbalanced sample sizes among settlements because a simulation study of GLMMs has shown remarkable robustness to unbalanced designs [4]. We have explicitly stated our reason for using GLMM in Line 194-195.

L137 The information about 'spot' size is redundant and can be deleted.

Response: We have deleted the redundant description (Line 149).

L149-153 It would be good to also use external information about the climatic niche. Please see general comment.

Response: As we explained above, we could not develop a database of the climatic niche of species because reliable distribution data of Asian butterflies outside Japan are not available. The correlation between climate and abandonment in our dataset was low and the risk of confounding would be negligible for our dataset.

L155 Change to: "We applied three ..."

Response: We have revised the text as suggested (Line 167).

L159 Please provide more information. Settlement should be nested within region. I guess you used each 'spot' as a data point, or did you aggregate the data to land use type? Please provide more information. Without aggregation, you need to include habitat type as additional random effect, nested within spot, nested within region.

Response: In the revised manuscript, we have provided information on the nested survey design and unit of analysis (Line 171-177). Because we treated the presence/absence of a species in a plot as the unit of analysis, we included habitat type as a random effect in the statistical analyses (Line 174-175). Fortunately, the results were almost identical to those in the previous version of the manuscript, and the conclusion of the study was not affected.

L165 Do you mean you included random slopes and intercepts?

Response: We considered random slopes of species, but we used species-specific fixed intercepts (i.e. a vague prior for each species, Line 215-216) because the distribution of intercepts was outside the scope of this research. We have added descriptions of these model settings (Line 181,198)

L170 You want to assess large-scale context specificity and I was wondering why you did not include interaction effects of abandonment and temperature. However, be aware that both variables are collinear which can cause wrong coefficient estimates (see general comment).

Response: We agree the interaction between temperature and land abandonment would also be very interesting. However, including the interaction term is not compatible with our research aim, namely, estimating species-specific sensitivities to land abandonment and testing their associations with climatic niche and habitat preference. This is because the main effects alone would not have ecological meaning if the interaction were considered [5]. We also revised the Introduction to clarify that our focus was species-specific sensitivity to land abandonment (Line 86-94).

L172 I would not consider the three land use types as confounding effects but rather as an interesting question itself. Does the impacts of abandonment differ among the land use types? You may test this with respective interaction effects.

Response: For the same reason why we did not consider the interaction between land abandonment and temperature, it was difficult to include the interaction of land use type and land abandonment simultaneously with habitat preference.

L179 What about 'paddy'?

Response: Paddy field was the baseline category of the dummy variable for land use. We have noted this in the text (Line 189).

L200 Unclear structure of the sentence.

Response: We have revised the sentence for clarity (Line 216-218).

L219 For me it is not clear how you keep the information about abandonment if you replace it with 'trait' information. Or did you change the response variable using the response to abandonment (respective coefficient from Eqn. 1)?

Response: We apologize for this unclear sentence. We actually modelled the average effect of abandonment $\mu_{\beta 1}$ as a function of the habitat preference of each species (Line 245). Then, coefficients in eqn. 2 (α_1 and α_2) and species-specific coefficients of land abandonment were estimated simultaneously.

L290 Change to "... the upper 95% CI ..."

Response: We have revised this as suggested (Line 292).

L300 From Fig. 3 it seems that forest species do not respond at all.

Response: You are right. The effects of land abandonment on forest species differed significantly from other habitat types but was nearly equal to zero. We have revised the sentence to note this (Line 301-303).

L310 Does this mean that 33 species responded positively?

Response: Yes, the posterior means of 37 and 33 species were negative and positive, respectively. The posterior means of 7 species were negative in the upper 95% CIs. We have added a detailed description on Line 309-311.

L337 "species pools"

Response: We have corrected the spelling (Line 344).

L338 "regions"

Response: We have corrected this (Line 345).

L339 "reflect the historical"

Response: We have added "the" (Line 346).

L349 Please reorder the sentence from abandonment to herbs to hosts to openland butterflies.

Response: We reordered the sentence as suggested (Line 359-361)

L353 Good point, but please be more clear to explain that habitat degradation is quite immediate for openland butterflies while positive effects for forest species take much longer.

Response: We have added a statement noting that habitat degradation for openland butterflies occurred quickly, while forest butterflies would need a longer period of time to experience a positive effect (Line 357-359).

L363 How did you show and test this?

Response: It was difficult to specify the reason why the effects of local land use were rarely significant, and we noticed that this paragraph was tangential to the main point of our discussion. Therefore, we have removed the paragraph from the revised manuscript.

Fig. 4 Are CL and CG absolute numbers? Please provide proportions instead, which is more informative since it relates to the total number of species predicted in a grid cell.

Response: We have added the proportions of CL and CG for all the species in Fig. 4c,d and for Red List species in Fig. S3c,d.

References:

1. Ohwaki A. 2018 How should we view temperate semi-natural grasslands? Insights from butterflies in Japan. *Glob. Ecol. Conserv.* **16**. (doi:10.1016/J.GECCO.2018.E00482)
2. Wesche K, Ambarlı D, Kamp J, Török P, Treiber J, Dengler J. 2016 The Palearctic steppe biome: a new synthesis. *Biodivers. Conserv.* **25**, 2197–2231. (doi:10.1007/s10531-016-1214-7)
3. Schweiger O, Harpke A, Wiemers M, Settele J. 2014 CLIMBER: Climatic niche characteristics of the butterflies in Europe. *Zookeys* **367**, 65. (doi:10.3897/ZOOKEYS.367.6185)
4. Schielzeth H *et al.* 2020 Robustness of linear mixed-effects models to violations of distributional assumptions. *Methods Ecol. Evol.* **11**, 1141–1152. (doi:10.1111/2041-210X.13434)
5. Engqvist L. 2005 The mistreatment of covariate interaction terms in linear model analyses of behavioural and evolutionary ecology studies. *Anim. Behav.* **70**, 967–971. (doi:10.1016/J.ANBEHAV.2005.01.016)

Appendix B

Response letter

We appreciated the acceptance of our paper. We corrected the manuscript following all the suggestions by Dr. Schweiger (Line 57-58, 238, 265, 270, 379, 412 and 422). We also added the recent publications relevant to our study in the references (Line 68, 83 and 415). These publications have no impact on the significance and originality of our study. We changed the reviewer-only URL of data repository (Dryad) to the URL for publication (Line 459).